# Phonon Bridge Effect in Superlattices of Thermoelectric TiNiSn/HfNiSn With Controlled Interface Intermixing

**DOI:** 10.3390/nano10061239

**Published:** 2020-06-25

**Authors:** Sven Heinz, Emigdio Chavez Angel, Maximilian Trapp, Hans-Joachim Kleebe, Gerhard Jakob

**Affiliations:** 1Institute of Physics, Johannes Gutenberg University, Staudingerweg 7, 55128 Mainz, Germany; jakob@uni-mainz.de; 2Graduate School of Excellence Materials Science in Mainz, Staudingerweg 9, 55128 Mainz, Germany; 3Catalan Institute of Nanoscience and Nanotechnology (ICN2), CSIC and The Barcelona Institute of Science and Technology (BIST), Campus UAB, Bellaterra, 08193 Barcelona, Spain; emigdio.chavez@icn2.cat (E.C.A.); kleebe@geo.tu-darmstadt.de (H.-J.K.); 4Institute of Applied Geosciences, Darmstadt University of Technology, Schnittspahnstrasse 9, 64287 Darmstadt, Germany; trapp@geo.tu-darmstadt.de

**Keywords:** interface, thermal conductivity, superlattice, intermixing, coherent phonon, roughness, 3 omega, 3 omega method, magnetron sputtering, half-Heusler, thermoelectric, thin film, TiNiSn, HfNiSn, thermal boundary resistance

## Abstract

The implementation of thermal barriers in thermoelectric materials improves their power conversion rates effectively. For this purpose, material boundaries are utilized and manipulated to affect phonon transmissivity. Specifically, interface intermixing and topography represents a useful but complex parameter for thermal transport modification. This study investigates epitaxial thin film multilayers, so called superlattices (SL), of TiNiSn/HfNiSn, both with pristine and purposefully deteriorated interfaces. High-resolution transmission electron microscopy and X-ray diffractometry are used to characterize their structural properties in detail. A differential 3ω-method probes their thermal resistivity. The thermal resistivity reaches a maximum for an intermediate interface quality and decreases again for higher boundary layer intermixing. For boundaries with the lowest interface quality, the interface thermal resistance is reduced by 23% compared to a pristine SL. While an uptake of diffuse scattering likely explains the initial deterioration of thermal transport, we propose a phonon bridge interpretation for the lowered thermal resistivity of the interfaces beyond a critical intermixing. In this picture, the locally reduced acoustic contrast of the less defined boundary acts as a mediator that promotes phonon transition.

## 1. Introduction

Thermoelectric generators convert heat to electricity without the need for moving parts, making them particularly maintenance efficient [1]. In transportation and energy production they can scavange waste heat, which otherwise accounts for 60% of primary energy expenditure [2]. Here, more effective thermal barrier materials maintain a larger heat gradient, thus improving their conversion rate. Therefore, a variety of techniques of thermal conductivity engineering are explored, such as composite generation [3], layer structure design [4] and nano-structuring [5]. As a common element in many of these techniques, additional interfaces partly impede heat flow [6]. For an appropriate choice of materials, they block a broad band of lattice vibrations [7], while electronic properties are maintained. However, the complex relationship between interface structure and thermal properties challenges both experimental and theoretical researchers. Depending on the materials vibrational spectra and interface topography, phonons scatter diffusely, thus randomizing their direction, reflect or transmit at an interface [8]. The ratio of these mechanisms determines the thermal resistance and is highly specific to material, temperature and growth conditions. Superlattices provide a valuable framework to study these effects with respect to their interface properties. They consist of a periodic arrangement of epitaxial layers that form a coherent crystal structure. They represent an effective design of thermal barriers for a wide variety of materials and are heavily interface dominated systems. Therefore any change of interface quality is expected to have a decisive effect on overall thermal properties.

In this study, TiNiSn/HfNiSn acts as a model system, because both constituents are important thermoelectric materials [9,10] and exhibit a high growth compatibility towards each other [11]. As Ti and Hf are isoelectronic, the combined TiNiSn/HfNiSn system exhibits little interface-related electronic perturbation [12]. The small electronic contribution to the heat transport in these semiconductors therefore does not depend on the superlattice structure [13]. However, as Hf and Ti have significantly different atomic masses, the interfaces of the composite potentially act as effective thermal barriers [14]. Additionally, because they are closely related compounds, they form a coherent crystal structure in cross-plane direction [15].

## 2. Materials and Methods

The stoichiometric sputter targets have been prepared by arc melting the constituent metals, coarse graining of the arc melted balls, and spark plasma sintering of 50 mm diameter targets in vacuum. As TiNiSn/HfNiSn shows little intrinsic intermixing, the pristine superlattice exhibits sharp interfaces [13]. An additional deposition step of a mixed stoichiometry Ti${}_{0.5}$Hf${}_{0.5}$NiSn interlayer at the interface simulates increased intermixing, thus emulating a system with less defined intrinsic interfaces and otherwise similar properties. Figure 1 illustrates the structure of two exemplary sample sets of 2 by 2 and 6 by 6 unit cell (UC) superlattices, respectively. For both superlattice periods, it shows three examples, in which different amounts of artificial intermixing by Ti${}_{0.5}$Hf${}_{0.5}$NiSn deposition are illustrated.

The growth properties of these half Heusler thin films depend on the magnetron sputter deposition conditions [15], reaching an optimum for 5 Pa of pressure during sputtering, a deposition temperature of 445 °C, a sputtering current of 60 mA at a voltage of around 250 V, and a target substrate distance of 4 cm. The crystalline quality is monitored using a Bruker D8 Discover X-ray diffractometer in Bragg-Brentano geometry with a copper anode.

The 2θ-θ-diffraction patterns indicate the growth characteristics. In order to quantify crystalline quality, rocking curves of the peaks observed in θ/2θ scans were measured [16]. In a rocking curve the 2θ angle, i.e., the length of the scattering vector corresponding to the lattice constant, is fixed, while ω is varied around the θ value of the peak. Thus only the direction of the momentum transfer is varied and probes the orientational distribution of the lattice planes, i.e., the crystalline grains. The widths of the rocking curves therefore serve as indicators for the crystalline quality. The individual grains of the studied thin film systems consist of crystallites with lateral dimensions around ∼100 nm and typically extend over the whole film thickness. In a previous study [14] the thermal properties were shown to depend critically on the crystal quality in general and on the rocking curve width specifically. As a systematic drift of crystal quality over the sample series could distort the observed effects, the rocking curve is monitored over the whole sample series. In this way, it can be verified that the change of thermal properties is caused by interface-related effects and not by a trivial change in growth characteristics.

Additionally, the superlattice parameters are monitored by recording characteristic satellite peaks at specific distance from main film peaks. The superlattice period *d* and the X-ray wavelength λ determine the satellite angle θ [17]:(1)$$\frac{2\sin(\theta )}{\lambda }=\frac{1}{\delta }\pm \frac{n}{d}.$$

Here, $\delta =\frac{\delta_{A}+\delta_{B}}{2}$ is the mean diffraction plane distance and *n* the satellite order. While the spacings of the satellites reveal the superlattice period *d*, their intensities indicate the interface definition [17]. At the lowest period length an attenuation of satellite-peaks intensity can be clearly observed with increasing degree of tailored intermixing. However, the effect is negligible for longer period lengths due to the small extension of the intermixing layer compared to the superlattice period length.

The thickness of the artificial intermixing layer that is deposited in the additional interlayer deposition step is determined from a closely monitored deposition rate and a controlled deposition time.

This intermixing layer thickness is independently verified by high angle annular dark field high-resolution scanning transmission electron microscopy (HAADF HR-STEM), performed on a JEOL JEM ARM 200F equipped with a Cs-corrected condenser system, operated at an acceleration voltage of 200 keV.

After the morphological characterization, the nanostructures were investigated with regard to their thermal conductivity κ (i.e., thermal resistivity ρ=1/κ), using the 3ω-method [18,19]. For this purpose, a heater was structured in a lithography step on top of an insulating layer consisting of a combination of AlO${}_{x}$, MgO and SiO${}_{2}$. The heater corresponds to a 4-terminal set-up, in which an AC-heating current $I=I_{0}\sin\omega t$ is applied, while the voltage response is monitored at the same time by separate contacts. By analyzing the voltage response to the current, the thermal characteristics of the system can be inferred, as heating effects lead to additional higher harmonics in the signal. The cause of these higher harmonics can be understood by analyzing the oscillating temperature rise *T* caused by the heating power *P*:(2)$$\begin{array}{cc} P=I{\left(t\right)}^{2}R & =\frac{1}{2}I_{0}R^{2}\left[1-\sin\left(2\omega t\right)\right] \end{array}$$(3)$$\begin{array}{cc} \Rightarrow T & =\hat{T}\sin\left(2\omega t\right) \end{array}$$
with the source current frequency ω and the amplitude of the temperature oscillations $\hat{T}$.

Because the resistances of the heater is temperature dependent, this oscillating temperature leads to a resistance oscillation proportional to the temperature coefficient $\alpha =\frac{1}{R_{0}}\frac{dR}{dT}$:(4)$$\begin{array}{c} R=R_{0}\left[1+\alpha \hat{T}\sin\left(2\omega t\right)\right]. \end{array}$$

From this oscillating heater resistance the higher harmonic contribution to the voltage signal with a 3ω characteristic follows directly from Ohm’s law:(5)U=RI(6)=R0I0[1+αT^sin(2ωt)]sin(ωt)(7)=U0sin(ωt)+12αT^sin(3ωt)−sin(ωt)

With the 1ω-voltage component $U_{\omega }=U_{0}\left(1-\frac{1}{2}\alpha \hat{T}\right)$ and the 3ω-component $U_{3\omega }=\frac{1}{2}U_{0}\alpha \hat{T}$. As $\frac{1}{2}\alpha \hat{T}\ll 1$, the temperature oscillations can be inferred from the comparison of the harmonic components and the known temperature coefficient of the heater material:(8)$$\begin{array}{c} \frac{2U_{3\omega }}{\alpha U_{\omega }}\approx \frac{2U_{3\omega }}{\alpha U_{0}}=\hat{T} \end{array}$$

These temperature oscillations predominantly depend on the properties of the substrate. However, for a heater that is much wider than the overall film thickness, the film causes a constant offset that is proportional to its thermal resistance. As illustrated in Figure 2, the heat flow in this case is quasi 1-dimensional in the cross-plane direction of the thin film and edge-effects can be neglected. By comparing the temperature oscillations of a thin film sample ${\hat{T}}_{f}$ with a reference sample ${\hat{T}}_{R}$, the thermal resistance of the film $R_{f}$ can be extracted from this offset [20]:(9)$$\begin{array}{c} {\hat{T}}_{f}-{\hat{T}}_{R}=\frac{P}{l\times 2b}\left[R_{f}+\left(R_{aux,f}-R_{aux,R}\right)\right] \end{array}$$
where *P* corresponds to the applied electrical power, *l* to the heater length, *b* to the heater half-width and $R_{f}$ to the thermal resistance of the film. $R_{aux}$ expresses the thermal resistance that is supplied by the auxiliary layers, the insulating layer and a 20 nm vanadium and 20 nm TiNiSn buffer layers that has been inserted to improve crystal growth. The auxiliary layers are reproduced as close as possible in the reference sample to ensure comparability to the sample of interest, so that $(R_{aux,f}-R_{aux,R})\approx 0$. To counteract the effect of target aging and the associated drift in thermal properties, multiple reference samples are used to verify insulating layer uniformity across the sample series. Additionally, the samples were produced in a random order with respect to their period length.

To account for the variation in the geometrical factors *b* and *l*, the heaters are characterized with an optical microscope for every sample separately and Equation (Equation 9) is adapted accordingly. The overall reproducibility is verified with the repetition of several measurements on separate samples. For this purpose, double sample holders were used to grow thin films at identical sputter conditions. The resulting variation suggests an error of the used method of ∼6.5%.

The determined thermal resistivities of the superlattices were analyzed by separating a bulk-like and an interface contribution, where the bulk like contribution corresponds to the constituent materials weighted by their material content
(10)$$\rho_{bulk}=\frac{\rho_{1}d_{1}+\rho_{Int.}d_{Int.}+\rho_{2}d_{2}}{d_{1}+d_{Int.}+d_{2}}.$$
where the reference thermal conductivities κ=1/ρ were measured on 1 μm thick layers of TiNiSn, $1/\rho_{1}=$ 6.17 W/mK, HfNiSn, $1/\rho_{2}=$ 2.76 W/mK, and Ti${}_{0.5}$Hf${}_{0.5}$NiSn, $1/\rho_{Int}=$ 1.97 W/mK. Superlattices exhibit an excess resistivity additional to this bulk-like term, which is given by the interface thermal resistances and by partial confinement within the superlattice layers:(11)$$\rho -\rho_{bulk}=\rho_{intf}=\phi +\frac{R_{TBR}}{d}$$
with the finite size effect term ϕ and the thermal boundary resistance $R_{TBR}$. $R_{TBR}$ corresponds to the resistance caused by incomplete transmission of heat carrying phonons across the interface, while the size effect term expresses the increase in thermal resistivity caused by phonon confinement. The ratio of interface and bulk-like contribution is illustrated by the interface thickness equivalent:(12)$$d_{equi}=\frac{\rho_{intf}}{\rho_{bulk}}d,$$
which gives the amount of material that corresponds to the thermal resistance contribution of a single set of interfaces.

To estimate the expected interface contribution to thermal resistance we follow the analysis of Alvarez et al. [8], which combines a treatment within the framework of the Diffusive Mismatch Model (DMM) [6] with the Acoustic Mismatch Model (AMM) [21]. Here, AMM describes atomically flat interfaces appropriately, while DMM is applicable to irregular interfaces with high interface scattering rates. The combination allows to identify trends, compare samples with different interface qualities and to estimate the order of magnitude of the thermal boundary resistance. With the degree of interface definition quantified by the specularity parameter p= [0,1], the transmission function is given by:(13)$$\begin{array}{c} \begin{array}{ccccc} \Gamma & = & p\Gamma_{S} & + & \left(1-p\right)\Gamma_{D} \end{array} \end{array}$$
where $\Gamma_{S}$ and $\Gamma_{D}$ are the angular integrated transmission coefficients of AMM [21] and DMM [6], respectively, estimated using the material parameters shown in Table 1.

Given the transmission function, an effective device size is defined, which quantifies the partial confinement within the layers [8]:(14)$$\begin{array}{c} L_{eff}=\left(1-\Gamma \right)L_{1}+\Gamma \left(1-\Gamma \right)L_{2}+\ldots +\left(1-\Gamma \right)\Gamma^{N-2}L_{N-1}+\Gamma^{N-1}L_{N} \end{array}$$
where $L_{n}=nL$, with the individual layer size *L*, and *N* as the total number of individual layers. As each period contains one layer of TiNiSn and one layer of HfNiSn, the thickness of a single layer is half the period length, L=d/2. The term (1−Γ) is the share of phonons that are confined within one layer and Γ(1−Γ) is the share that are transmitted to the next layer and are confined there. From this effective length, the finite size term in Equation (Equation 11) can be calculated following the extended irreversible thermodynamics framework [23]:(15)$$\begin{array}{c} \phi =\rho_{bulk}\left(\frac{2\pi l^{2}}{L_{eff}^{2}}{\left[\sqrt{1+{\left(\frac{2\pi l}{L_{eff}}\right)}^{2}}-1\right]}^{-1}-1\right), \end{array}$$
here the mean free path *l* is estimated from the bulk thermal conductivities according to the Debye-Callaway model [24,25].

The second part of Equation (Equation 11), the thermal boundary resistance, is also a function of the transmission coefficient. It can be approximated for a transition from material *i* to material *j* by [6,26]:(16)$$\begin{array}{c} R_{ij}=\frac{4\pi^{2}\hbar^{3}v_{i}}{k_{B}\Gamma_{ij}T^{3}}\frac{1}{\int_{0}^{\theta_{D}/T}x^{4}e^{x}/{\left(e^{x}-1\right)}^{2}dx}, \end{array}$$
with the Debye-temperature $\theta_{D}$, the Debye-velocity $v_{i}$, the Boltzmann-constant $k_{B}$ and the reduced Planck-constant *ℏ*. From the equations above the mean thermal boundary resistance is derived as
(17)$$\begin{array}{c} R_{TBR}=\frac{R_{ij}+R_{ji}}{2}-\rho_{i}l_{i}-\rho_{j}l_{j}, \end{array}$$
where contributions from layers adjacent to the interface, $\rho_{i}l_{i}$ and $\rho_{j}l_{j}$, are substracted, which are implicitly included in Equation (Equation 16).

## 3. Results

Main diffraction peaks confirm the (002)-growth direction with a rocking curve width around 0.6–1.0°, demonstrating consistent, epitaxial growth.

Figure 3 shows a XRD-pattern around the (002) main film peak of a 6 by 6 superlattice with 1 UC artificial intermixing. The fit by a homemade algorithm [27] extracts the period length. Additionally, for the smallest period length, a significant reduction in satellite intensity is visible. However, for longer period lengths this effect diminishes, as the intermixing layer goes down to only ∼12% of total layer volume.

The increased intermixing is separately validated in HR-STEM images. Figure 4 shows two samples, one pristine SL and a SL with a unit cell (UC) of an added Ti${}_{0.5}$Hf${}_{0.5}$NiSn interlayer. Both samples show clearly defined layers through the majority of the sample volume. The natural intermixing in pristine superlattices is typically around 0.3 nm, which can be estimated from a previous study [28]. In the second sample, Ti${}_{0.5}$Hf${}_{0.5}$NiSn has been deposited at every interface that corresponds to 0.6 nm leading to an effective broadening of the intermixing layer. Both predicted values are displayed as overlays in the corresponding images and agree well with the apparent broadening of the boundary layers. This verifies that the presented method produces SL model systems with similar period length and differing degrees of (artificial) intermixing.

Figure 5a displays their thermal resistivity ρ=1/κ (Km/W) as a function of period length *d*. Three sets of symbols indicate Ti${}_{0.5}$Hf${}_{0.5}$NiSn interlayers of 0, 0.5 and 1 UC, i.e., 0, 0.3 nm and 0.6 nm, respectively. The blue bars show the respective bulk resistivities of the constituent materials that were previously determined. While for long periods, the thermal resistivity corresponds to the mean of the bulk resistivities, it increases significantly for lower periods, i.e., higher interface densities. This increase demonstrates the dominance of interface and superlattice effects in this regime. The solid lines represent a fit based on
(18)$$\begin{array}{c} \rho =\rho_{bulk}\frac{\left(d+d_{equi}\right)}{d} \end{array}$$
where the material equivalent $d_{equi}$ serves as the adjustable parameter. The fit extracts the effective contribution of a single set of interfaces to the thermal resistance as
(19)$$\begin{array}{c} R_{intf}=\rho_{bulk}d_{equi} \end{array}$$

In Figure 5b both interface and bulk-like contribution are displayed separately with the point $d=d_{equi}$ being marked. At this point, interface and bulk-like effects contribute equally. In (c) the rocking curve-width of the measured samples is given as a proxy for crystal quality, demonstrating consistent growth over the sample series.

The extracted material equivalents and the corresponding effective thermal interface resistances are given in Table 2. In the half unit cell superlattices, a single interface corresponds to 3.8 nm additional material, which is 27% more compared to a pristine superlattice. In contrast, further increase of intermixing leads to material equivalent of only 2.3 nm, and thus a decrease of 23% as compared to the same reference.

## 4. Discussion

The effective interface thermal resistance is directly correlated with the transmission function Γ of phonons across an interface.

The theoretical transmission function, evaluated by Equation (Equation 13), is shown in Figure 6a for different values of *p*. As in most materials, the transmission is higher in the AMM-case. However, for this system the acoustic contrast is small compared to common model systems like Si-Ge or AlAs-GaAs, resulting in a transmission of over 95%.

The finite size terms for the fully specular and fully diffusive case are given in Figure 6b,c, respectively. They express the partial confinement by the incomplete transmission of phonons across the interfaces and are based on Equation (Equation 15) using the previously calculated Γ. For the diffusive case, the maximum value of ϕ reaches 1.5, which corresponds to an additional 150% of bulk thermal resistivity resulting from phonon confinement. For a period length of 10 nm, the size term drops to a modification of only ∼9%. In the AMM case the modification does not exceed 4%, which is the result of the high transmission rate and the subsequent inefficient confinement.

The confinement effect together with the thermal boundary resistances give the effective interface thermal resistance per superlattice period
(20)$$\begin{array}{c} R_{intf}=2R_{TBR}+\phi d, \end{array}$$
which is given in Figure 7, separated into the distinct contributions.

In Figure 7a the thermal boundary resistance is shown, which is significantly higher in the diffusive ($3.9\times 10^{-9}$ Km${}^{2}$/W) than in the specular case ($0.3\times 10^{-9}$ Km${}^{2}$/W). As stated above, the same holds true for influence of the finite size term. Its effective contribution to the thermal resistance of a period, ϕd, is given in Figure 7b. As this value depends on the period length, three values are shown, a value corresponding to d= 1 nm, one for d= 10 nm and a value averaged over all period lengths. However, the contribution is roughly 2.5 times smaller than the TBR in the diffusive case and 17 times smaller in the specular case. Consequently, the thermal resistance of the superlattices are strongly dominated by the TBR-term (Figure 7c).

All investigated systems exhibit a low effective interface resistance, which indicates a high specularity and thus a comparatively high interface quality. Relative to the reference superlattice with no additional intermixing, the sample with a 0.5 UC interlayer exhibits a higher thermal resistance, which is expected for a lower interface definition. However, the 1 UC interlayer samples show lower thermal resistances, despite of the markedly lower interface quality revealed in the TEM-study. This effect is likely caused by the finite extension of the boundary layer, which can be comparable to the main layer thickness for small period length. Several non equilibrium molecular dynamics suggest that the intermixing layer, which naturally exhibits intermediate acoustic properties, can serve as a buffer [29,30], mediating phonon transmission. This mediating effect competes with the initial lowering of interface specularity by a thin intermixing layer. Consequently Yang et al. [31] find that moderate interface intermixing maximizes thermal resistivity in the simulation of a model superlattice, in agreement with our findings.

## 5. Conclusions

We have successfully manufactured thin film superlattice model systems with tunable intermixing and period length. An artificial intermixing layer modifies the interface quality for parts of the sample series: A 1:1 mixture of the main components deposited at the interface effectively extends the intermixing boundary layer. The thermal resistance was measured by a 3ω-method and analyzed by separating the interface and bulk-like contributions. The expected thermal boundary resistances are estimated using an analytical model based on a combination of acoustic mismatch model and diffusive mismatch model. We find that thin intermixing layers increase interfacial thermal resistance significantly, which is most likely caused by an uptake in diffuse scattering. However, for strongly increased intermixing, the thermal resistance decreases. As an explanation, a locally reduced acoustic contrast by the intermixing is considered, promoting interface transition of phonons [30].

We conclude that in nanostructures both effects have to be considered, especially for characteristic sizes lower than 2–4 nm, where interface effects dominate thermal transport.

## Figures and Tables

**Figure 1 nanomaterials-10-01239-f001:**
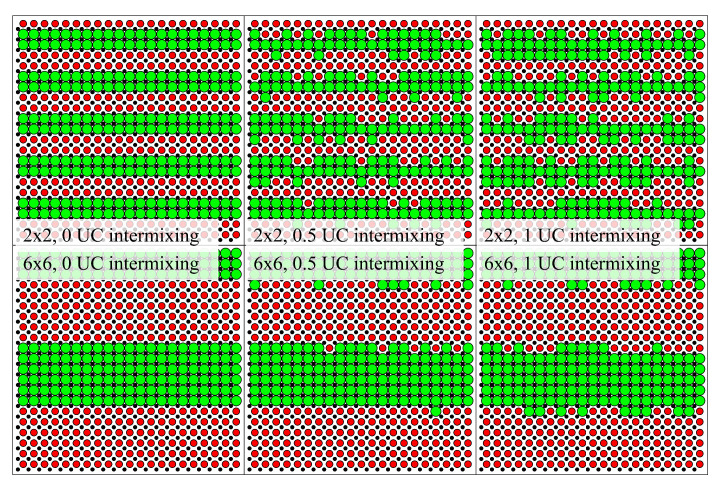
Schematic representation of the investigated TiNiSn/HfNiSn superlattice designs. Black dots correspond to the common elements Ni and Sn, while Ti and Hf are symbolized by red and green dots, respectively. The annotation gives the superlattice period in units cells (UC) and the amount of artificial intermixing.

**Figure 2 nanomaterials-10-01239-f002:**
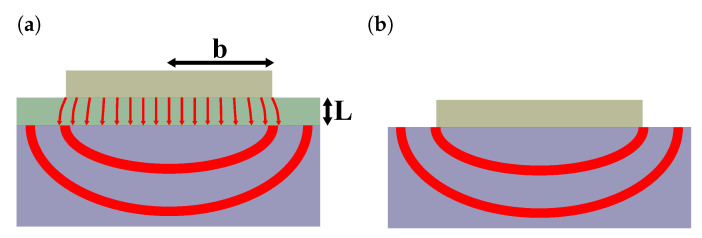
Heat flow caused by the heater structure for a substrate with a thin film (**a**) and for a bare substrate (**b**). As the heater is much broader than the overall film thickness the heat flow is quasi 1-dimensional and the edge effects are negligible.

**Figure 3 nanomaterials-10-01239-f003:**
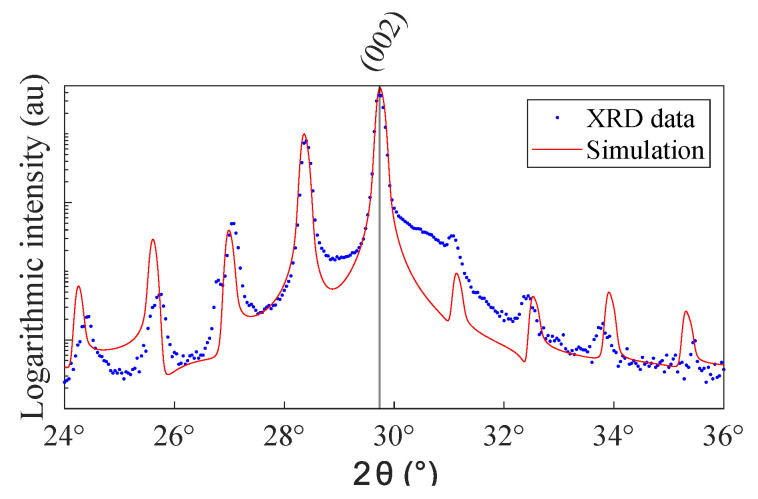
Typical XRD diffraction pattern in the vicinity of the (002)-peak. The investigated sample is a 6 by 6 UC superlattice with 1 UC artificial intermixing. The red lines correspond to a simulation using a homemade algorithm [27].

**Figure 4 nanomaterials-10-01239-f004:**
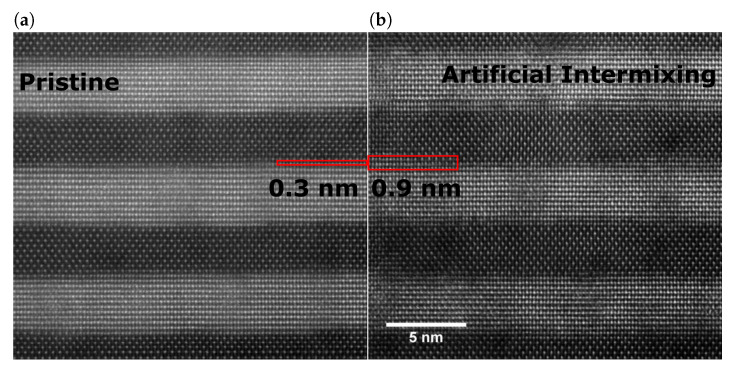
HR-STEM images of superlattice samples with differing degrees of artificial intermixing: (**a**) Pristine superlattice with only intrinsic intermixing. (**b**) Superlattice with an added intermixing of one unit cell, i.e., 0.6 nm.

**Figure 5 nanomaterials-10-01239-f005:**
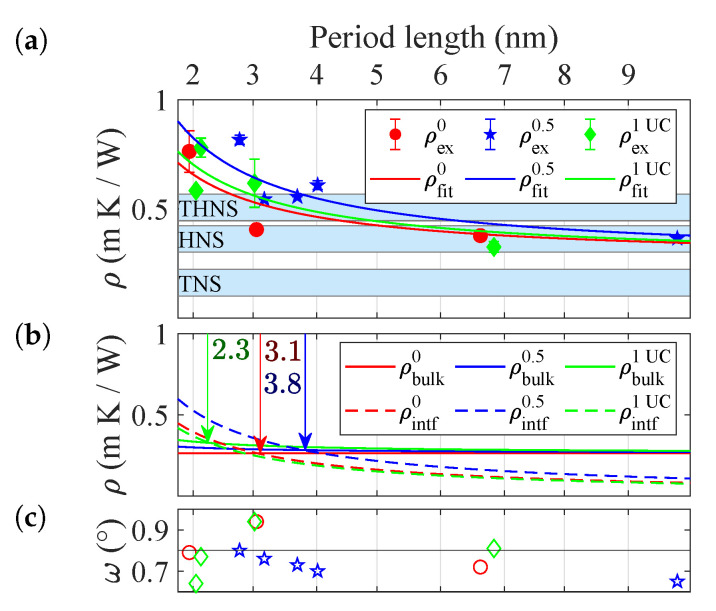
Thermal resistivities $\rho^{\alpha }=1/\kappa$ of superlattices of different period length and Ti${}_{0.5}$Hf${}_{0.5}$NiSn interlayer thickness of α= 0, 0.5 and 1 unit cells (UC), respectively. (**a**) Experimental $\rho_{ex}^{\alpha }$ thermal resistivities together with the fit $\rho_{fit}^{\alpha }$. The blue bars indicate the thin film thermal resistivies of the constituent materials TiNiSn, HfNiSn, and Ti${}_{0.5}$Hf${}_{0.5}$NiSn. (**b**) Breakdown of interfacial $\rho_{intf}^{\alpha }$ and bulk-like $\rho_{bulk}^{\alpha }$ contribution to the thermal resistivity. The point of equal contribution is marked with an arrow. The bulk-like contributions correspond to a weighted average of the bulk resistivities of the constituent materials. (**c**) The rocking curve widths ω as a proxy for crystal quality.

**Figure 6 nanomaterials-10-01239-f006:**
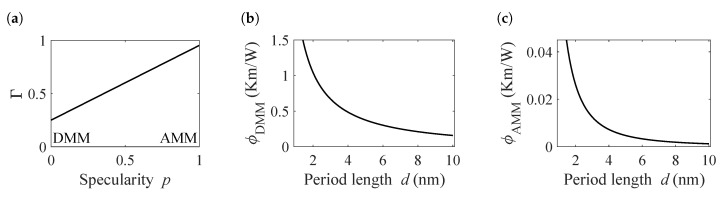
Overview over the transmission function and the finite size terms for different values of the period length *d* and specularity parameter. (**a**) The transmission function in dependence on the specularity parameter. p= 0 corresponds to a fully diffusive interface interaction, which is described by the diffusive mismatch model (DMM). The fully specular case is described by the acoustic mismatch model (AMM). (**b**) Finite size term in the DMM case, which quantifies the modification of the bulk thermal resistivity by phonon confinement effects. (**c**) Finite size term in the AMM case.

**Figure 7 nanomaterials-10-01239-f007:**
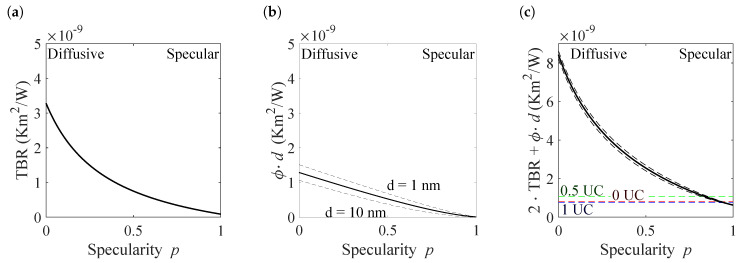
Contributions to the effective interface thermal resistance. (**a**) The thermal boundary resistance describes the effect of incomplete transmission of heat-carrying phonons across an interface. (**b**) The influence of the finite size term per superlattice period. As the finite size term depends on the period length, two values for 1 and 10 nm are given together with the mean value. (**c**) The total effective interface thermal resistance per superlattice period. The experimental values are marked for superlattices with intermixing layers of 0, 0.5, and 1 unit cells (UC), i.e., 0, 0.3 and 0.6 nm.

**Table 1 nanomaterials-10-01239-t001:** Literature values used in the estimation of the effective interface thermal resistance. The Debye velocity has been calculated from the band structure simulations given in [12].

	Debye Velocity $v_{d}$ (m/s)	Heat Capacity $C_{i}$ ($\times 10^{6}$ J/m${}^{3}$K)	Debye Temperature $\theta_{D}$(K)
Source	[12]	[22]	[22]
TiNiSn	3560	2.31	380.1
HfNiSn	3090	2.65	316.2

**Table 2 nanomaterials-10-01239-t002:** Material equivalents for different interface designs. The material equivalent expresses how much additional material corresponds to an individual interface in terms of thermal resistance.

Artificial Intermixing Layer Thickness (Unit Cells)	Interface Material Equivalent $d_{\mathrm{equi}}$ (nm)	Effective Thermal Interface Resistance ($\times 10^{-9}$ Km${}^{2}$/W)
0	3.0	0.81
0.5	3.8	1.08
1	2.3	0.75

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
