# Peer review of "Phonon Bridge Effect in Superlattices of Thermoelectric TiNiSn/HfNiSn With Controlled Interface Intermixing"

_nanomaterials, 2020, doi:10.3390/nano10061239_

Round 1

Reviewer 1 Report

The authors have significantly improved their manuscript since

the last version, therefore I recommend the publication of the manuscript.

Author Response

The authors thank the referee for the valuable input.

Reviewer 2 Report

The authors have revised the desired comments in the revised version

Author Response

(The authors gave the same response as above.)

Reviewer 3 Report

Heinz et al present an experimental study on the effects of the interface on the thermal conductivity of multilayers. This is relevant for thermoelectric materials, among others (lower the better). They study multilayers of half Heusler alloys in which they can control to a certain degree the quality of the interface by tuning the parameters of the magnetron sputtering. Their experimental finding is that the lowest thermal conductivity can be assigned to the multilayer with intermediate quality. 

Most of the manuscript is dedicated to a detailed analysis of the results, based on a large number of referenced works. The measured thermal conductivities are portioned off and assigned to various parts of the system, to the bulk of the constituent layers, to the boundary effects, etc., with the aim to tease out the effect of the quality of the interface. This is certainly rather interesting and important.

I think the manuscript should be published, but I make a few comments:

The analysis is based on the 3w measurements of thermal conductivity, yet scant description of this is given, only a reference. I don't think this Journal places limits on word-count, so for completeness, the measurements ought to be described better. How accurate is this measurement? How does its error compare to the differences studied between samples? The measurement is not trivial, since it is based on a comparison (difference) between a film and a bare substrate, there is an additional, undescribed, insulating kayer, the geometry of the deposited heater probably also plays a role. All of these are relevant for the study.

The same sort of comment holds for the treatment of the x-ray reflectance. This is a crucial piece experimental information, and it is apparently analysed to a certain degree, but this is not described, only a reference to a homecooked algorithm is given. I am surprised that quantitative information about the interface quality is not extracted, either by this algorithm, or using software like SUPREX.

And the same comment holds for the STEM study: is the interface roughness quantified from the images? How?

Eventually, the message of Fig.4 seems to be that the chief way to reduce thermal conductivity is to make the constituent layers very thin, the interface quality doesn't add much variation for the same thickness. Doesn't this overarching effect defy the purpose of the study?

Furthermore, Fig.6c seems to say, that the interface roughness obtained here has a trivial effect on the thermal conductivity, what would really help woiuld be to decrease the specularity.

I don't think that the Conclusion reflects properly some of the (unappealing) findings.

I have a problem with the Introduction, too. I am aware that it is hard to introduce a paper on thermoelectrics in any novel way, since so many papers have been publsihed in the last decade. Nevertheless, talking about climate change in light of the nuanced interest of the manuscript seems far fetched. However, what is missing is a description of how a thin-film or multilayer thermoelectric would be used. This is jarring, since a very detailed formula on the efficiency is given. Most such papers conform themselves with putting zT=S^2*sigma*T/kappa which would be plenty for this paper with its focus on reducing kappa. However, if the Introduction is geared to give more details into thermoelectrics, then the Reader would apperciate a few thoughts on how this could work? The thermal conductivity is measured transverse to the film. Would the Seebeck effect also be measured this way? And the electrical conductivity? Would the temperature gradient be established across the few nm thick film? Is that reasonable? I understand that the manuscript does not aim for a full thermoelectric characterization, but the gap between climate change and a complete lack of ideas on the to-be-employed geometry is jarring. 

In addition, there are a few curious expressions (they may be OK, but I find them strange):

"reflex" instead of reflection?

line 47: in dependence of?

Author Response

The authors thank the referee for the valuable input that was especially helpful to
improve the description of the used methods.

In the attached document the review report of Referee 3 will be cited with the response given in the indented paragraphs. Excerpts of the original and the revised manuscript are given with the changes being highlighted and marked with line and page numbers. 

This manuscript is a resubmission of an earlier submission. The following is a list of the peer review reports and author responses from that submission.

Round 1

Reviewer 1 Report

The article: “Phonon Bridge Effect in Superlattices of Thermoelectric TiNiSn/HfNiSn With Controlled Interface Intermixing” describes the changes of thermal properties on thin film superlattice model systems with tunable intermixing and period length. Enhanced scattering processes of phonons are reported for strongly increased intermixing. It is shown that the thermal resistance decreases because the intermixing layer acts as an acoustic buffer. The measurement techniques used in this manuscript have been selected very well and their applicability is presented for these studies. The result obtained are sound and clear. The paper is well written and organized.

However, it is desirable that the authors consider revising Figure 4. This illustration is very overloaded and the results are difficult to track. The reviewer had enlarged the illustration significantly to see the details (bars, dotted lines, dashed lines). Perhaps the presentation of the results in several diagrams would make more sense. The reviewer is looking forward to the final paper.

Reviewer 2 Report

The paper is interesting, because authors managed to make layered

structure of Heusler alloys, and check whether its thermal conductivity

(which is important for the Figure of Merit ZT) changes as expected on the

basis of simple model of serial and parallel connections. These experimental study was certainly not easy to accomplish and because of this I recommend

publish of the paper after the authors address following question:

How the thermal conductivity was measured in detail. The 3w technique

as I understand can measure the time constant of the RC circuit related

to heat capacity and heat conductivity. How, were the absolute values

of thermal resistivity determined shown in Fig 4. What heat capacity was assumed?

Reviewer 3 Report

The paper by Heinz et al. reports on thermal transport measurements in TiNiSn/HfNiSn with a tight control on the thickness of the interface intermixing region.

I have found this work very difficult to evaluate. On the one hand the control demonstrated on the synthesis of the superlattices and on their intermixing is really remarkable and paves the way to a systematic study of heat transport in this kind of systems; on the other hand the authors used an oversimplified sophomoric model to interpret the data and their analysis (11 manuscript lines) provides a very poor insight in the underlying physics.

As, in my opinion, documenting a technical advance does not guarantee by itself publication Nanomaterials, I regret to conclude that my recommendation is rejection of this contribution.